# Needle Penetration Simulation: Influence of Penetration Angle and Sample Stress on the Mechanical Behaviors of Polymers Applying a Cast Silicone and a 3D-Printed Resin

**DOI:** 10.3390/ma15165575

**Published:** 2022-08-13

**Authors:** Thore von Steuben, Florian K. Enzmann, Sebastian Spintzyk, Frank Rupp, Ariadne Roehler

**Affiliations:** 1Department of Vascular Surgery, Medical University Innsbruck, Anichstraße 35, 6020 Innsbruck, Austria; 2Medical Materials Science and Technology, University Hospital Tuebingen, Osianderstraße 2-8, 72076 Tuebingen, Germany; 3ADMiRE Lab–Additive Manufacturing, Intelligent Robotics, Sensors and Engineering, School of Engineering and IT, Carinthia University of Applied Sciences, Europastraße 4, 9524 Villach, Austria

**Keywords:** needle penetration, 3D printing, mechanical properties, material testing, injection angle, vessel wall stress, material testing, surgical training, training models, cannula angle

## Abstract

For surgical catheterization training applications, realistic and effective materials are desired. In this study, the relevance of a needle puncture angle and a simulated wall stress on different elastic materials were determined in a previously developed experimental setup. Both settings were considered individually in two new setups. In addition, a control setup with neither angle nor prestress was designed. During the process of puncturing the samples of two materials (Replisil 9N and Formlabs Elastic 50A), force–displacement values were collected, and three predefined parameters evaluated. The differences between the angled/stressed groups and the control group were analyzed. The additively processed material required a significantly higher force to puncture than the conventional one (*p* < 0.001). Moreover, a needle angulation of 45° required more force than puncturing orthogonally. Prestressing the samples did not clearly influence the resulting force. An evaluation of relative parameters showed that the investigated materials behaved differently but not linearly differently under the influence of needle angle and prestress. Therefore, it is essential to evaluate the properties and suitability of materials for surgical training models in appropriate experimental setups considering multiple parameters.

## 1. Introduction

Clinical practice models have a high potential for improving a patient’s outcome in emergency situations and other critical conditions [1]. In addition, surgeons benefit from modern training models to learn or prepare techniques without risks or chance of harming patients [2]. Instead of an elaborate fabrication of a mold for castable materials, 3D printing is widely known for its rapid and more accessible manufacturing of complex shaped models and is also applicable for models of the individual inner anatomy of living patients. For this purpose, data from computed tomography and magnetic resonance imaging are used to create digital 3D models [3]. These models can be printed and used for surgical guidance and teaching purposes. There are many methods for the technical implementation of 3D printing. Whereas all of them manufacture a workpiece layer by layer, the mechanisms of linking the layers differ. The structural integrity of the printed layers is important for precision and therefore the insufficient hardness of soft materials is a limitation to printing devices [4]. Printable materials have become more versatile, so that complex and specific applications are realizable [5]. For training purposes, softer materials have enabled 3D printing to compete with castable polymers, but they lack the structural integrity a mold can provide [6]. Of course, additively processed materials must also meet the requirements necessary for realistic and effective surgical training. This requires a detailed characterization of novel materials and tissues concerning their suitability prior to their application in training models.

To evaluate the suitability of materials for vessel puncture models, a setup was created in preliminary work [7]. The needle penetration test reported in this case enabled to examine nonbiological materials for their puncture properties and to compare them with biological samples. In that test setup, adjustable penetration angles were realized, and natural occurring wall stress of given vessels could be transferred to the simulating sample.

Although specific angles of insertion for injections or catheterizations are recommended in the literature [8], little is known about their influence on the mechanical behavior of the tissue. It is also not known how varying wall stress of different vessels may affect surgical catheterization.

The developed experimental setups of that research were used to investigate the isolated influence of the penetration angle and the simulation of wall stress (further denominated as “prestress”). Since in the previous study all specimens were measured with a prestress and an angled needle, it was decided to perform a measurement at 45° without prestress, one at a 90° orientation with prestress and one at 90° without prestress as the control group. As materials to be investigated, one castable two-component silicone and an additively processed polymer were chosen, which proved to differ significantly in maximum penetration force in a previous study [9].

The aim of this study was to support the development of a test rig which examines the suitability of different materials for simulating vessel walls in surgical catheterization training. For this purpose, the influence of prestress and cannula penetration angle on the mechanical behavior was investigated in materials with different manufacturing methods (cast, 3D-printed) and Shore hardnesses. A changing mechanical behavior in specific situations will have an impact on the selection of materials for many applications, e.g., training models or vessel prosthetics.

## 2. Materials and Methods

### 2.1. Conceptualization of the Test Setups

To isolate the penetration angle and the prestress from the previous setup, three adjustments had to be made, details of which are described in the following sections. The digital 3D models of the attachments were created using the computer-aided design (CAD) software program Siemens NX Student Edition 1926 (Siemens, Munich, Germany), exported as standard tessellation language (STL) files. With Prusa Slicer Software (version 2.3.3, Prusa Research a.s., Prague, Czech Republic) the STL files were converted into the g-code format and then printed. All solid components, except for screws and nuts, were additively manufactured with the fused filament fabrication (FFF) printer Prusa i3 MK3S (Prusa Research a.s., Prague, Czech Republic) and a polylactide filament (PLA) (Prusa Research a.s., Prague, Czech Republic). Multiple models were printed at the same time on the same device. All parts were printed in the quality preset with a layer thickness of 0.15 mm. The support structures were generated automatically during slicing, the extrusion temperature was set to 215 °C, the heat bed temperature to 60 °C and the infill to 30%. For all 3D-printed parts, the STL files are provided in the Appendix A.

#### 2.1.1. Parts from the Preliminary Work

Some parts were adopted from a previous study [7], e.g., the needle holder, which clamps a specific cannula type (see Appendix C) in place with a constant orientation of its angled opening. The cannula was clamped at its shaft with a polylactide cuboid and a M6 screw countered with a nut embedded in the holder; this resulted in a constant alignment with the test direction. For the horizontal alignment of the setup, an adjustment module was implemented at the base of the testing machine. The orientation could be locked in place with three M6 screws. Moreover, the method for prestressing the samples with weights, described in more detail in Section 2.1.3, was adopted.

#### 2.1.2. Cubic Adapter

First, a cubic adapter with a socket as counterpart was designed for anchoring all other new parts in the testing machine. The socket had to connect the circular socket of the testing machine with the cubic one on the attachments. This preserved the constant orientation between the sample and attachment changes.

#### 2.1.3. Setup Attachment for 90° and Prestress

The principle of this attachment was adopted from the preliminary setup [7]. Instead of a square plateau, a circular one was designed (Figure 1a). The bottom of this plateau was mounted on the above-mentioned cubic adapter. On the top, the plate had sections, which divided the circle in eight slices with the same size, to guide the clamps (Figure 1b–e). Those clamps were also adopted and served as socket for the samples. Each of them consisted of a base (Figure 1b,c) and a lid (Figure 1d,e), which was screwed together with an M6 screw countered with a nut clamped in the base (Figure 1c) and placed on the circular plateau. Strings were attached to all clamps and threaded through holes in the side of the plate (Figure 1a). To prestress the samples and thereby simulate a vessel wall stress, weights were attached at the end of the strings. Each clamp–string pair had a custom weight (Table 1). The dimensions of a common thoracic aorta (diameter of 33 mm, wall thickness 2 mm) served to calculate the axial and tangential wall stress. Then, the force and the resulting masses were calculated from the wall stress and the cross-section area of the sample dimensions. The calculations and results were taken from the preliminary work [7].

#### 2.1.4. Setup Attachment for 45° without Prestress

To penetrate the samples at a fixed angle of 45° without any prestress, a new sample holder was designed (Figure 2). The cubic adapter on the bottom prevented any rotation and a constant orientation of the sample to the cannula’s opening was ensured (Figure 2a,c). Onto the cubic adapter, an angled socket for the circular samples with a diameter of 20 mm was placed (Figure 2b,c). The depth of the socket was 1 mm, which was half the thickness of the samples (Figure 2c). On the outside of this socket a thread was designed (Figure 2a–c). Together with the lid (Figure 2d,e), screwed to the thread, the sample was clamped in the holder. Since the angulation limited the passage for the cannula, a bevel was added to the lid (Figure 2e).

#### 2.1.5. Setup Attachment for 90° and No Prestress

A 90° penetration angle without prestress was aimed as a control group for the other results. Therefore, a neutral flat mount was designed (Figure 3a). Here, as in the previously described attachment, the same cubic adapter was used at the bottom and the same specimen holder (Figure 3a). The socket for the sample was also the same as in the attachment for 45° and no prestress. Solely, this new sample holder had no angulation in between, so it resulted in a 90° angle. The lid was the same as for the attachment for 45° and no prestress (Figure 3b).

### 2.2. Manufacturing of the Samples

As one of the materials to be tested, Replisil 9N (duplicating silicone, Silconic^®^, Lonsee, Germany) was chosen. It is a two-component castable polymer with a low modulus of elasticity and a Shore hardness of 9A. It was already tested by Salewski et al., as the softest material in their examination [9]. In order to obtain 115 mm × 115 mm × 2 mm sample mats, a mold was created. The mold was designed with Siemens NX Student Edition 1926 (Siemens, Munich, Germany) and printed with the vat polymerization printer Form 3B (Formlabs, Somerville, MA, USA) from Grey resin (Formlabs, Somerville, MA, USA). The mold consisted of two parts, one part with ventilation shafts and a lid, which was screwed in place with seven M6 screws countered by nuts on the other side. The STL files are provided in the Appendix A. The two components were mixed in a 1:1 ratio with the aid of single use plastic measurement cups and plastic stirrers (see Appendix C) and poured into the mold while standing on a vibrating plate for minimal bubble formation. After 20 min of hardening, the vibrating plate was turned off, the screws were loosened, and the mat was taken out. The mold and the lid were cleaned with isopropanol (SAV LP GmbH, Flintsbach, Germany) after every cast. 

A large excess of sample mats were created to give the option of sorting out defective ones. To create evenly shaped circular samples with a diameter of 20 mm, a punch (DIN 7200, Gedore toolfactory GmbH & Co. KG, Remscheid, Germany) was used. 

As additively processed material, the Formlabs Elastic 50A (Formlabs, Somerville MA, United States) was chosen. The Elastic 50A is a photopolymer with a Shore hardness of 50A. It was the hardest material tested by Salewski et al. [9]. First, a grid with 5 × 5 cubic holes with dimensions of 23 mm × 23 mm × 25 mm each and a wall thickness of 2 mm was designed with Siemens NX Student Edition 1926 (Siemens, Munich, Germany) (Appendix D). This design was chosen because the walls acted as their own support structures and a maximum number of samples could be obtained with only one print. The grid was also printed with the Form 3B and processed following the manufacturer’s recommendations. Initially, the model was washed in the Form Wash (Formlabs, Somerville, MA, United States) by filling with isopropanol (Gatt-Koller GmbH, Absam, Austria) for 10 min on the platform and additionally 10 min without platform. Then, the model was cured in the Form Cure (Formlabs, Somerville, MA, USA) for 20 min at 60 °C. Finally, a large number of samples were punched out of the grid walls with the same punch mentioned above to have enough excess for sorting out defective ones.

Each sample was visually inspected and checked for air inclusions. Replisil 9N samples with visible air inclusions larger than ca. 0.5 mm were sorted out and not included into the test series. Visibly deformed Formlabs Elastic 50A samples were also eliminated. After defective samples had been sorted out, *n* = 75 samples for each material were passed on to the tests.

### 2.3. Conduction of the Tests

For the tests, the Zwick Z010 (ZwickRoell GmbH & Co. KG, Ulm, Germany) universal testing machine was used with a load cell with a maximum force of 500 N. For the setup assembly, first, the adjustment module was mounted at the bottom socket of the machine. On top of it, the cubic adapter was placed. After ensuring the linear alignment of all parts, the sample was placed in the corresponding sample holder of the specific attachment and mounted in the cubic adapter. On the upper socket, the needle holder was placed. A fresh special cannula REF 219.14 1.4 × 70 mm (Vygon Erzeugnisse für Medizin und Chirurgie GmbH & Co. KG, Aachen, Germany) was placed in the needle holder, the test procedure was started and the force with its affiliated position was recorded (Figure 4). A cannula speed of 1 mm/min and no termination criteria were set. The detailed settings can be found in the Appendix B. For each measurement, a fresh cannula was used. After the defective samples had been disposed of, *n* = 25 samples were tested of each material (Replisil 9N, Formlabs Elastic 50A) and setup (90° prestress, 90° no prestress, 45° no prestress) combination. This resulted in a total tested sample count of *n* = 150.

### 2.4. Data Evaluation

During the measurements, force and displacement were recorded. Three parameters were chosen for further analysis. The gradient of the linear part at the beginning of each curve (E), the first force peak (F_1_) and the second force peak (F_2_) (Figure 5).

All calculations and analyses were performed with Excel (version 2203, Microsoft Corporation, Redmond, DC, USA). E was determined by the Excel formula SLOPE, while F_1_ and F_2_ were determined by the Excel formula MAXIMUM. The mean and standard deviation (SD) were calculated. A Mann–Whitney U test was performed for the two materials for all three parameters of the three setups (α = 0.05). Additionally, each result of the Replisil 9N was subtracted from each result of the Formlabs Elastic 50A for all three parameters in the according test series, resulting in a 25 × 25 Matrix and 625 differences. The data were processed this way because changes in those differences would indicate that the materials behaved differently under specific conditions. Again, the mean and SD were calculated and a Mann–Whitney U test was performed for each value between the series with 90° without prestress, 45° with prestress and the control group 90° without prestress, respectively (α = 0.05). A flow chart of the data evaluation is given in Appendix D (Figure A1).

## 3. Results

### 3.1. Condition of the Samples

The Replisil 9N samples were mostly homogeneous. Small air inclusions were visible (Figure 6). The force–displacement diagrams had similar shapes in all three test series. The Formlabs Elastic 50A samples were homogeneous in their core but showed parallel grooves on the surface (Figure 6). Due to the designed locking mechanism of the holders for tests without prestress, some samples bent upwards when closing the lid. Then, the examiner loosened the lid until no bend was visually observable.

### 3.2. Graph Analysis and Measurement Results

As the traverse relative starting point for the cannula was set at the beginning of each series and some samples bent more upwards than others, sometimes the measurement recording did not start at 0.00 N. This did not affect the measurement results as the slope and maximum forces where not dependent on the starting force.

The characteristic graph areas (Figure 5) were observable for all materials in all test series. The graph’s starting point represented the cannula’s penetration of the most upper layer. The penetration of the bottom surface with the tip of the cannula was represented by the first force peak F_1_. The progression between the starting point and F_1_ was linear. Only few graphs showed a small peak in between. Afterwards, the force either dropped rapidly, as for most Formlabs Elastic 50A samples, or the force decreased almost linearly, as for most Replisil 9N samples. Either way, this drop in force showed the completed passage of the cannula’s tip. Till the second peak, F_2_, the force increased again. After the entire cutting edge of the cannula had passed through the bottom layer of the sample, the force dropped and showed an approximately constant value as long as the rest of the cannula pushed through the sample. This represented the friction of the outer cannula surface inside the sample body. The force–displacement graphs within the Replisil 9N group had similar and smooth progressions. The graphs within the Formlabs Elastic 50A group differed after F_1_ and seemed distorted (see Figure 7). The resulting means and SDs are listed in Table 2 and shown in Figure 8. The values of all parameters in all groups were significantly (*p* < 0.001) lower for the Replisil 9N than the Formlabs Elastic 50A.

### 3.3. Results from Calculations

The calculated differences between the two materials are shown in Table 3 and Figure 9. In comparison with the control group (90° no prestress), the ∆E and ∆F_1_ values for the 45° no-prestress measurement were significantly lower (*p* < 0.001). ∆F_2_ showed no significant difference for 45° no prestress (*p* = 0.237). The measurement difference with 90° with prestress was significantly lower for E (*p* < 0.001). ∆F_1_ and ∆F_2_, on the other hand, were significantly higher compared to 90° no prestress (*p* < 0.001).

## 4. Discussion

The differences in penetration force of the prestressed samples significantly increased in both peak forces, but differed vastly in their magnitude, whereas the differences in elasticity decreased significantly. For the angled samples, the differences in elasticity and the first peak decreased significantly, while the second peak force showed no significant change at all. The predicted changes in the mechanical behavior upon penetration with an angle or prestress were evident in the experiments, as the examined values changed independently. Therefore, the influence of those parameters is important to consider in material comparison for penetration simulation in surgical training models. The results substantiating these claims are discussed below. During the test, it was observed that the needle sometimes slid abruptly through the additively manufactured samples because it got stuck on the groove structure caused by the production process. It is assumed that this was also the reason for the force deflections in the diagrams of the Elastic 50A, which, however, never influenced the average course of the curves.

In general, more force was necessary to penetrate the additively processed material Elastic 50A than the two-component silicone Replisil 9N. This phenomenon was already expected as the shore hardness of these materials differed clearly according to manufacturer’s specifications. Including the results of Salewski et al. [9] as an angled and prestressed group (60° with prestress) for both materials, the mean values were higher than those of the 90° with prestress. This also occurred between the groups at 45° without prestress and 90° without prestress. It can therefore be concluded that an angled insertion requires a higher penetration force. Comparing the groups in terms of their prestress (60° with prestress vs. 45° no prestress, 90° with prestress vs. 90° no prestress), no such pattern emerged. One might therefore assume that the prestress had no influence on the penetration forces.

However, considering the calculated differences between the materials it became obvious that the materials behaved differently but not linearly differently under the influence of angle and prestress. In the groups without prestressing (45° and 90° no prestress), ∆E and ∆F_1_ behaved significantly differently with respect to each other, while ∆F_2_ did not differ significantly. With an angulation of 45°, ∆E was only minimally lower compared to the horizontally clamped group, while ∆F_1_ was almost 65% lower. All values increased after putting an angle to the sample. In F_1_, the increase in force was higher for the Replisil 9N than for the Elastic 50A. As expected, the diagonal path through the angled sample was always longer than the straight one, which explains the increase in all parameters. Additionally, the elastic deformation in the vertical direction changed the overall tensile stress at 90° to a combination of tensile and pressure stress. This affected the more elastic Replisil 9N more than the stiffer Elastic 50A.

An isolated change in prestress (both 90° groups) showed a significant difference for all three parameters, but whereas both forces ∆F_1_ and ∆F_2_ were higher for the prestress group, ∆E was lower compared to the control group. The extent of difference between ∆F_2_ of 90° no prestress and 90° with prestress was also higher than for ∆F_1_. The decrease in ∆E with the simultaneous increase in ∆F_1_ and even higher in ∆F_2_ indicated an asymmetric influence on the material differentiation. The absolute values behaved differently between the two materials. The increase in E for both materials can be explained by the stretching of the sample. It decreased the elastic potential of the material beforehand and so steepened the slope. The increase in E can also be explained by the effect of entropy elasticity. The straightened and aligned polymer chains decreased the elasticity and strengthened the prestressed materials [10]. The chains tried to return to their natural random state and therefore created a retractive force, which acted as a counter-bearing. At the same time, the sample thinned under stress, which decreased the penetration force, which could be seen in F_1_. Both effects countered each other. Although, the stiffer material thinned less under stress, it was outmatched by the strong entropy elasticity effect, which resulted in the increase in F_2_ for the Elastic 50A.

Although sufficient and precise data for this evaluation were gained, improvements should be considered in further tests. The alignment and force measurements could be more precisely conducted by manufacturing the setup out of aluminum or steel. With adjustable springs instead of weights, different stress states would be possible, while providing exact forces on the clamps. The mold for the samples in the test set-up could be adjusted in a way that the samples would not bulge under more constant conditions. A greater variety of different angles or different stress states could also show if the change in differences between methods follow a specific trend.

Suitability tests of different nature have rarely been reported in other studies, but commonly with more simplified experimental setups compared to the purpose of our study [11]. An attempt for a needle penetration test was made by Lipton et al., by evaluating different two-layer materials with the penetration force of a sharp and a blunt needle [12]. Although they can be a first indicator for conceptualization, they are unable to represent the complex stress conditions and penetration angles in surgical interventions. In contrast, the novel setup presented in our previous study is capable to consider those circumstances and can also be applied to multilayer materials.

Further investigations of other factors could be of interest in the future. The influence of thickness or the wetting of the samples according to the suitability of materials for surgical practice could have an impact on material use for training models. As the penetration speed was very slow in our study, the influence of the needle’s velocity on the penetration force could also be investigated for clinical applications.

In the case of future materials for surgical training, the development is proceeding in different directions. While manufacturers have tried to use 3D printers with materials of lower Shore A hardness, some studies have been seeking to modulate the mechanical properties of polymers through hybrid materials [13]. The goal is to create softer materials for precision manufacturing of anatomical surgical training models, because the current materials are often too stiff and cannot match real tissue [14]. Another problem our study did not address is multimaterial printing. Most printers are not able to print different materials layer by layer, especially in the case of soft materials. An approach was made by Jaksa et al., who designed such a printer for functional anatomic models [15]. A polylactide and a silicone material were successfully printed and, among others, the adhesion between the two materials was discussed. The authors suggested that different printing materials might be required to mimic various tissues. In that case, the adhesive strength between components should be considered as important as the mechanical properties of the applied materials, and both must be investigated.

In summary, many factors affect the behavior of elastic materials towards needle puncture, as desired in clinical training models. First, the processing of the materials significantly influences the characteristic properties. While artifacts such as air bubbles can occur with molded two-component plastics, uneven surfaces must be expected with the modern variant of additive manufacturing. Furthermore, our study highlights that before a specific material can be selected for a needle penetration simulation, the influence of angle and prestress must be determined in detail for each material. Mostly, the influence was asymmetric and more complex than expected originally. It should be considered that properties such as entropy elasticity do not occur in every biological tissue because of the structural architecture of cells and their interaction with other cells or the extracellular matrix. Ultimately, training models must consist of several different materials, and suitable polymers will probably be necessary to sufficiently mimic tissue. Here, additive manufacturing already offers great material options, but needs further improvements. The manufacturing method also allows, e.g., multilayer models, and in combination with clinical imaging, patient-specific surgical training models can be created. Progress in multimaterial printing is desirable.

Overall, the findings related to the consideration of angle and prestress in material testing for vessel punctation training has proved to be a significant contribution to the improvement of 3D printing for this purpose. Single-type base materials, multicomponent materials and human tissues can be compared in a single setup with respect to their differences in nature and penetration conditions for the desired intervention simulation.

## 5. Conclusions

The established experimental setup has proven to be suitable for a detailed characterization of novel elastic materials to be used for surgical training models. Moreover, the significant influence of the penetration angle and a simulation of the vessel wall stress on material behavior was shown. Neither synthetic materials nor biological tissues follow a predictable or calculable behavior when exposed to conditions such as those previously mentioned. It is therefore imperative to evaluate promising materials for surgical training models in a test setup such as the one suggested in our study. Furthermore, the investigation of material behavior can be used to review previous guidelines for surgical puncturing (angle, pressure) and to develop recommendations in everyday clinical practice for specific angles considering different wall tensions.

## Figures and Tables

**Figure 1 materials-15-05575-f001:**
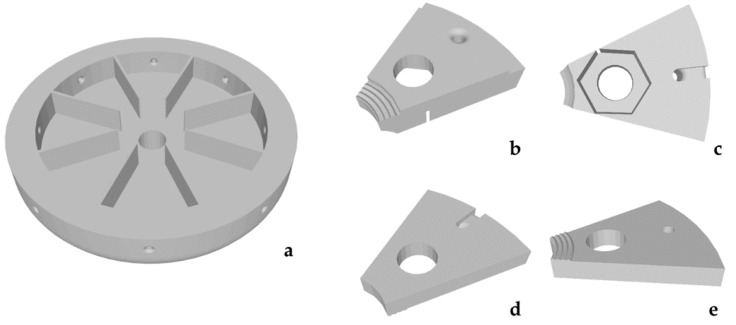
Setup attachment for tests at an angle of 90° with prestress. The plateau of the attachment viewed in an angled perspective from the top (**a**). The base of a clamp viewed from the top (**b**) and from the bottom (**c**). The lid of a clamp viewed from the top (**d**) and the bottom (**e**).

**Figure 2 materials-15-05575-f002:**
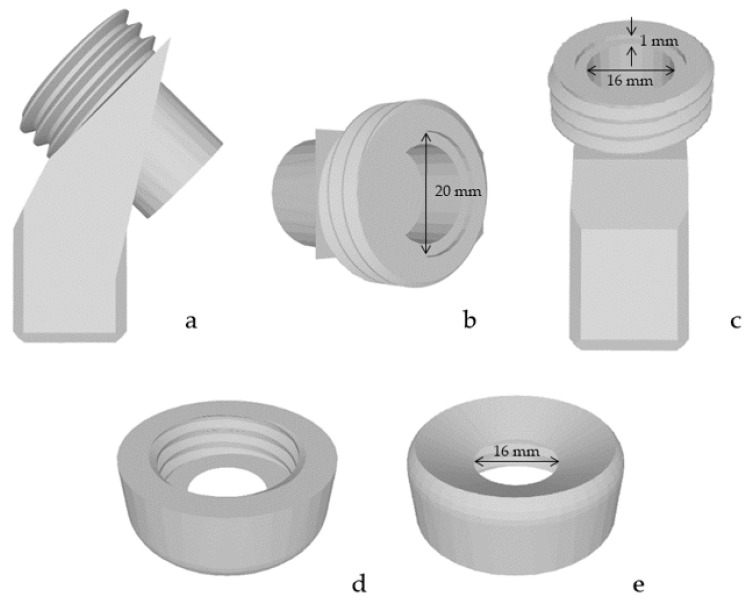
Setup attachment for tests at an angle of 45° without prestress. The base of the attachment with the cubic adapter viewed from the side (**a**), from the top (**b**) and the front (**c**). The lid of the attachment viewed angled from the bottom (**d**) and angled from the top (**e**).

**Figure 3 materials-15-05575-f003:**
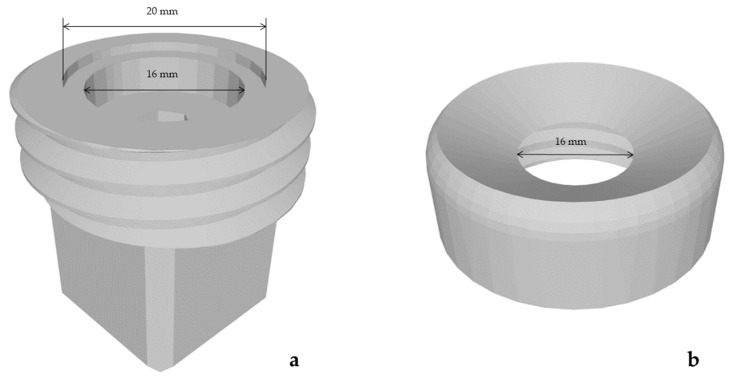
Set-up attachment for tests at an angle of 90° without prestress. The base of the attachment with the cubic adapter viewed in an angled perspective from the top (**a**). The lid of the attachment viewed angled from the top (**b**).

**Figure 4 materials-15-05575-f004:**
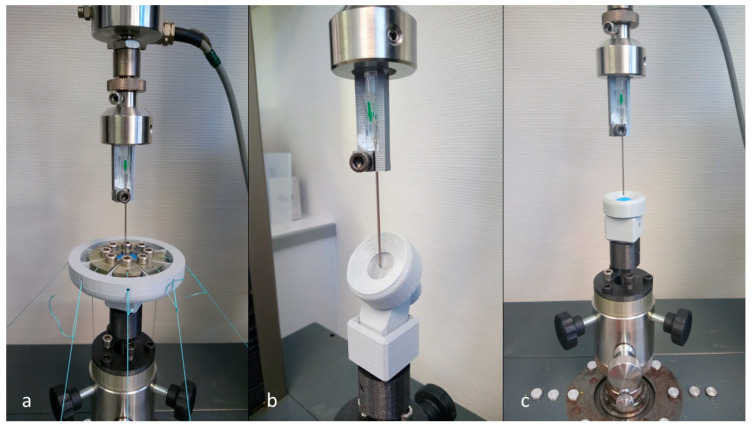
All three setups during the conduction of a test. The 90° prestressed setup testing a sample of Replisil 9N (**a**). The 45° non-prestressed setup testing a sample of the Formlabs Elastic 50A (**b**). The 90° non-prestressed control setup testing a sample of the Replisil 9N (**c**).

**Figure 5 materials-15-05575-f005:**
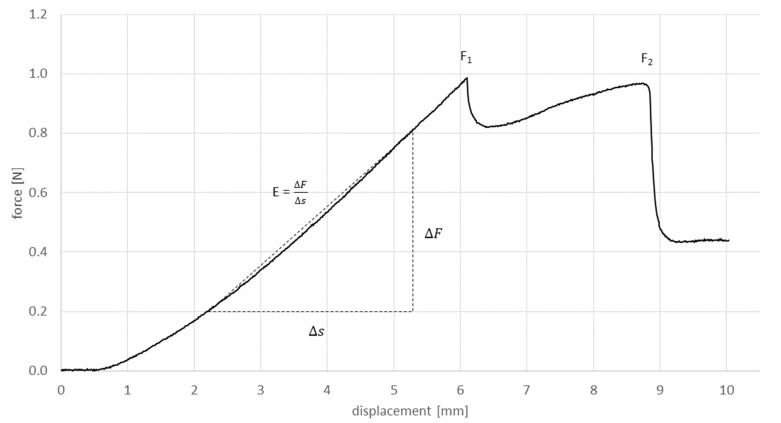
Schematic visualization of the evaluated values extracted from the force displacement graphs of the performed tests. E as the gradient of the change in force over the change in place. F_1_ as the first force peak and F_2_ as the following force maximum.

**Figure 6 materials-15-05575-f006:**
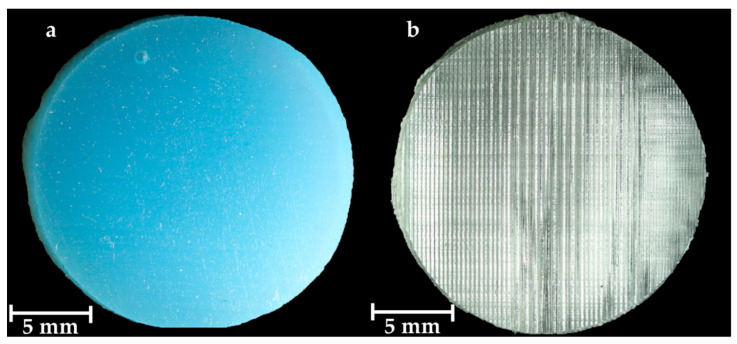
Pictures of both sample types. The Replisil 9N sample with a macroscopic air enclosure in the upper left quarter (**a**). The Formlabs Elastic 50A sample with vertical and parallel arranged grooves (**b**).

**Figure 7 materials-15-05575-f007:**
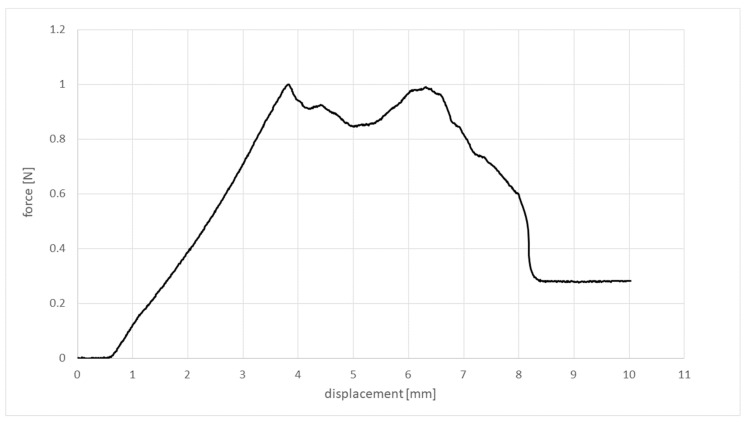
Force displacement diagram of one sample penetration at 90° without prestress. The force is presented in Newton and the displacement is presented in millimeters.

**Figure 8 materials-15-05575-f008:**
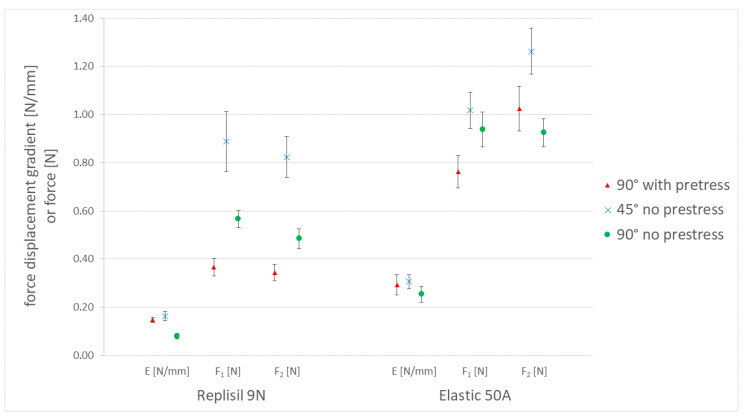
Visualization of the measured means and standard deviations of all three parameters, two materials and three groups. The slope E is in N/mm. The first force maximum F_1_ and the following maximum force F_2_ are in N.

**Figure 9 materials-15-05575-f009:**
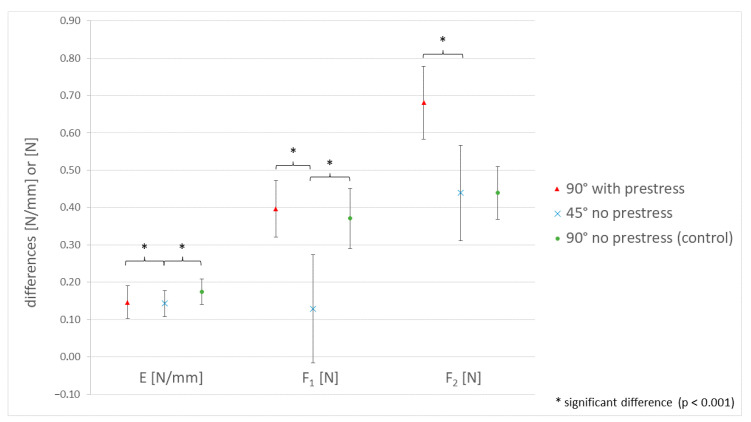
Visualization of the calculated means and standard deviations of the differences of all three parameters and groups. The gradient of the linear part ∆E is in N/mm. The gradient of the first force maximum ∆F_1_ and the gradient of the following maximum force ∆F_2_ are in N.

**Table 1 materials-15-05575-t001:** Masses of the weights for the attachment at 90° with prestress in gram, which were already calculated in our prior study [7]. The masses 1–8 are mentioned clockwise starting with the one on the opposite of the cannula’s opening.

	m_1_	m_2_	m_3_	m_4_	m_5_	m_6_	m_7_	m_8_
masses (g)	111.86	164.62	211.00	153.22	100.15	150.95	210.97	164.85

**Table 2 materials-15-05575-t002:** Mean test results of the three parameters E, F_1_ and F_2_ with standard deviations. The gradient of the linear graph part E is in N/mm. The first force maximum F_1_ and the following maximum force F_2_ are in N.

Test Series		E (N/mm)	F_1_ (N)	F_2_ (N)	*p* Value
90° with prestress	Replisil 9N (*n* = 25)	0.15 ± 0.01	0.37 ± 0.036	0.34 ± 0.035	<0.001
	Formlabs Elastic 50A (*n* = 25)	0.29 ± 0.042	0.76 ± 0.067	1.02 ± 0.092	
45° without prestress	Replisil 9N (*n* = 25)	0.16 ± 0.018	0.89 ± 0.124	0.82 ± 0.085	<0.001
	Formlabs Elastic 50A (*n* = 25)	0.31 ± 0.03	1.02 ± 0.075	1.26 ± 0.096	
90° without prestress (control group)	Replisil 9N (*n* = 25)	0.08 ± 0.012	0.57 ± 0.035	0.49 ± 0.041	<0.001
	Formlabs Elastic 50A (*n* = 25)	0.25 ± 0.032	0.94 ± 0.072	0.92 ± 0.058	

**Table 3 materials-15-05575-t003:** Mean test results of the differences of the three parameters ∆E, ∆F_1_ and ∆F_2_ with standard deviations. The gradient of the linear graph part ∆E is in N/mm. The first force maximum ∆F_1_ and the following maximum force ∆F_2_ are in N.

Test Series		∆E (N/mm)	∆F_1_ (N)	∆F_2_ (N)
90° with prestress	Formlabs Elastic 50A-Replisil 9N	0.15 ± 0.044	0.4 ± 0.076	0.68 ± 0.098
45° without prestress	Formlabs Elastic 50A-Replisil 9N	0.14 ± 0.038	0.13 ± 0.145	0.44 ± 0.128
90° without prestress (control group)	Formlabs Elastic 50A-Replisil 9N	0.17 ± 0.034	0.37 ± 0.08	0.44 ± 0.071

## Data Availability

Data are available in Appendix A. Raw data are available from the corresponding author upon reasonable request.

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
