# Peer review of "Needle Penetration Simulation: Influence of Penetration Angle and Sample Stress on the Mechanical Behaviors of Polymers Applying a Cast Silicone and a 3D-Printed Resin"

_materials, 2022, doi:10.3390/ma15165575_

Round 1

Reviewer 1 Report

Author in this study tested different materials for surgical training in terms of needle penetration. It presents interesting findings with results having clinical value. However, the manuscript will benefit from proof reading as it has a lot of grammatical errors. Specific comments are listed below for improvement:

1. Abstract, line 15: were to be determined-change to: were determined.

2. Introduction: computer tomography-computed tomography; got generalizable-became generalizable. The study purpose needs to be made clearer as readers would like to see the rationale of this study or what research gap will be addressed in this study.

3. Methods: details were well described. STL-should be defined as standard tessellation language.

4. Results:  fine. Overall, clearly presented.

5. Discussion: usually, the first paragraph is a summary of key findings, such as something like: This study shows that…. Please revise it accordingly.

Another weakness of your discussion is lack of comparing your results with other studies in the literature. Page 11, line 335-336, you cited the reference: An approach was made by Jaksa et al., who designed such a printer for func-336 tional anatomic models [13]. This needs to be explained more with regard to the findings of Jaksa et al.

Further you may need to consider highlighting the significant contribution of your study.

Study limitations are highlighted well.

Author Response

First of all, thank you for your constructive feedback to our paper! 
We ackknowlage your language correction and therefore send it to a graduated english corrector. We hope we could adress your other comments as you intended:

1.    C: "Abstract, line 15: were to be determined-change to: were determined."
       A: We changed the sentence as you recommended. Thank you.

2.    C: "Introduction: computer tomography-computed tomography; got generalizable-became generalizable. The study purpose needs to be made clearer as readers would like to see the rationale of this study or what research gap will be addressed in this study."
       A: Thank you for your feedback! We worked on the study aim in the introduction, to make the possible impact of our study clearer. 

3.    C: "Methods: details were well described. STL-should be defined as standard tessellation language."
       A: We described it as you recommended. Thank you.

4.    C: "Results:  fine. Overall, clearly presented."
       A: Thank you.

5.    C: "Discussion: usually, the first paragraph is a summary of key findings, such as something like: This study shows that…. Please revise it accordingly."
       A: We added a paragraph as you recommended. Thank you.

6.    C: "Another weakness of your discussion is lack of comparing your results with other studies in the literature. Page 11, line 335-336, you cited the reference: An approach was made by Jaksa et al., who designed such a printer for func-336 tional anatomic models [13]. This needs to be explained more with regard to the findings of Jaksa et al."
       A: We summarized the study of Jaksa et al. and put it in context of our study. Thank you for your comment!

7.    C: "Further you may need to consider highlighting the significant contribution of your study."
       A: We adressed the significance of our study at the end of the discussion. Thank you for your important feedback!

Thanks to you we could really improve the manuscript. We hope you share this opinion and looking forward to your review!

Reviewer 2 Report

The manuscript treats an interesting subject and explains systematically main objective of the project. Some recommendations are given below for the sake of the improvement of the manuscript:

   Introduction is too short and the details regarding the 3d printer and the process should have been mentioned there. Also, there is a lack of referencing.

·      It looks like the sentence was not finished in Line 67.

·       Check the grammar between the lines 89 to 91.

·       Assemblies can be given in Figure 1-2.

·       Line 142, “high elasticity” expression is quite vague, it can be changed with a better alternative.

·       A schematic view of how the tests were conducted should be given.

·       The caption of Figure 4 has not the same font.

·       It would have been better to give some images of the samples in sub-section 3.1.

·       Check Line 227 grammatically.

·       Line 281 therefor> therefore

·       It would be better to show a force equilibrium using a free body diagram during the needle penetration to better explain the effect of angle and the distance crossed over by the needle.

·       In the last paragraph of the discussion, the drawbacks of the tissue mimicking manufactured by single type base material were mentioned and multi-material 3D printers were proposed for a better precision. Instead of highlighting many times the drawbacks of your study, you should equally mention your advantages.

Author Response

First of all, thank you for your constructive feedback to our paper! 
We hope we could adress your other comments as you intended:

1.    C: "Introduction is too short and the details regarding the 3d printer and the process should have been mentioned there. Also, there is a lack of referencing."
       A: We further described the method of 3D printing in order to explain it for a broad audience and worked on the description of our aims. Thank you for this important feedback!

2.    C: "It looks like the sentence was not finished in Line 67."
       A: We changed the sentence as you recommended. Thank you.

3.    C: "Check the grammar between the lines 89 to 91."
       A: We checked the grammar as you recommended. Thank you.

4.    C: "Assemblies can be given in Figure 1-2."
       A: We implemented a new Figure (4), showing the three set-ups in action. Those also show their assembly while helping to understand the way the tests were conducted. Thank you for your important feedback!

5.    C: "Line 142, “high elasticity” expression is quite vague, it can be changed with a better alternative."
       A: We changed it to "low modulus of elasticity". Thank you for your comment.

6.    C: "A schematic view of how the tests were conducted should be given."
       A: We added a Figure and decided to move it to the appendix as we dont want to overload the paper with Figures. Thank you for you feedback!

7.    C: "The caption of Figure 4 has not the same font."
       A: We changed the Figure (now Figure 5) as you recommended. Thank you.

8.    C: "It would have been better to give some images of the samples in sub-section 3.1."
       A: We added a Figure (6) as you recommended. Thank you for this comment as it enhances this section!

9.    C: "Check Line 227 grammatically."
       A: We checked the grammar as you recommended. Thank you.

10.    C: "Line 281 therefor> therefore"
         A: We changed the spelling as you recommended. Thank you.

11.    C: "It would be better to show a force equilibrium using a free body diagram during the needle penetration to better explain the effect of angle and the distance crossed over by the needle."
         A: After a comprehensive discussion within our team, we think, that a static force equilibrium would not be correct as the samples itself are stretching while the needle penetration and alter the force directions. We added a Figure (4) with pictures of the set-ups in action. Together with the given explanations in the discussion the circumstances should be perspicuous to the reader. But thank you for your input!

12.    C: "In the last paragraph of the discussion, the drawbacks of the tissue mimicking manufactured by single type base material were mentioned and multi-material 3D printers were proposed for a better precision. Instead of highlighting many times the drawbacks of your study, you should equally mention your advantages."
         A: We elaborated on the discussion as you recommended and summerized the importance to all the mentioned aspects. Thank you for this important feedback!

Thanks to you we could really improve the manuscript. We hope you share this opinion and looking forward to your review!

Reviewer 3 Report

The manuscript reported by Steuben et al entitled as “Needle penetration simulation: Influence of penetration angle and sample stress on the mechanical behaviors of polymers applying a cast silicone and a 3D printed resin” attempts to development of a test rig which examines the suitability of different materials for simulating vessel walls in surgical catheterization training. The proposed idea is interesting and potentially could have good impact for surgical training models.  The results shown in the manuscript has clearly supported the claim of the study. Hence, in my opinion the manuscript can be accepted for publication after minor revisions. My specific comments about the manuscript are as follows.

1.      More details regarding the 3D printing process such as the extrusion temperature, dimension of the printed construct etc. should be provided in the materials and methods section.

2.      The developed 3D printed system should be compared to similar system currently available in literature. A good comparison with the existing methods/systems can compare and highlight the importance of the present system.

...........................................................................................................................................................

Author Response

First of all, thank you for your constructive feedback to our paper! 
We hope we could adress your other comments as you intended:

1.    C: "More details regarding the 3D printing process such as the extrusion temperature, dimension of the printed construct 
       etc. should be provided in the materials and methods section."
       A: The temperature for our print with the Prusa printer was added to the section. The Formlabs printer in which the samples 
       were printed just got few settings we completely described. Thank you for your important feedback!

2.    C: "The developed 3D printed system should be compared to similar system currently available in literature. A good 
       comparison with the existing methods/systems can compare and highlight the importance of the present system."
       A: We changed and expanded the particular part in the discussion. The literature shows a great lack of material comparison 
       for surgical practice models regarding their mechanical properties. We intend to fill this gap with our research for a 
       special application. Thank you for your feedback!

Thanks to you we could really improve the manuscript. We hope you share this opinion and looking forward to your review!

Round 2

Reviewer 1 Report

Authors have done a good job in revising the manuscript and all of my comments are addressed. Thank you!

Reviewer 2 Report

The authors improved their manuscript by considering the reviewers' recommendations. The manuscript is recommended for the publication after eliminating the journal template based discrepancies.